# Post-Wildfire Landslide Hazard Assessment: The Case of The 2017 Montagna Del Morrone Fire (Central Apennines, Italy)

**Cristiano Carabella**, **Enrico Miccadei ***, **Giorgio Paglia** and **Nicola Sciarra**

Department of Engineering and Geology, Università degli Studi dell'Adriatico "G. d'Annunzio" Chieti-Pescara, Viale Pindaro 42, 65127 Pescara, Italy; cristianocarabella12@gmail.com (C.C.); giorgiopaglia3@gmail.com (G.P.); nicola.sciarra@unich.it (N.S.)

*** Correspondence: enrico.miccadei@unich.it

**Abstract:** This work focused on a post-wildfire landslide hazard assessment, applied to the 2017 Montagna del Morrone fire. This wildfire increased the possibility of landslides triggering, as confirmed by the occurrence of a debris flow, triggered by an intense, short duration rainfall event in August 2018. The study area was investigated through a detailed analysis incorporating morphometric analysis of the topography and hydrography and geomorphological field mapping, followed by the landslide hazard assessment. In detail, the analysis was performed following a heuristic or expert-based approach, integrated using GIS technology. This approach led to the identification of five instability factors. These factors were analyzed for the construction of thematic maps. Hence, each factor was evaluated by assigning appropriate expert-based ranks and weights and combined in a geomorphology-based matrix, that defines four landslide hazard classes (low, moderate, high, and very high). Moreover, the morphometric analysis allowed us to recognize basins prone to debris flows, which, in relevant literature, are those that show a Melton ratio of >0.6 and a watershed length of <2.7 km. Finally, all the collected data were mapped through a cartographic and weighted overlay process in order to realize a new zonation of landslide hazard for the study area, which can be used in civil protection warning systems for the occurrence of landslides in mountainous forested environments.

**Keywords:** wildfire; landslide hazard; hazard assessment; geomorphology; debris flows; Central Italy

## 1. Introduction

Wildfire is a natural process in forest ecosystems and occurs with varying frequency and severity depending on landscape features and climatic conditions and is believed to have been relatively common since the late Devonian times [1]. It can be considered as a geomorphological agent, whose activity can vary enormously and depends on an interplay of factors including topography, geology, vegetation cover, fire characteristics, and rainfall patterns [2–4].

Landsliding is linked to the combination of geological, geomorphological, and climatic factors (instability factors) in response to trigger mechanisms, mostly represented by heavy rainfall events, seismicity, or human action [5–11]. Further predisposing conditions can be represented by wildfires, which induce significant changes in the landscape setting and give rise to post-wildfire hazards, such as erosion and sediment transport, floods, landslides (debris flows and rockfalls), and snow avalanches [12,13].

Landslide occurrence, in space or time, can be inferred from geomorphological hazard mapping and heuristic investigations computed through the analysis of environmental information or inferred from

geotechnical or physical-based models [14–22]. Hence, it is necessary to integrate and adapt the general knowledge, which is commonly based on the field assessment of geological and geomorphological settings, and slope stability to the analysis of post-wildfire situations, which is the first step in preventing further post-fire disasters [23].

Furthermore, wildfire can damage the vegetation cover acting on its structure and properties in three primary ways: By scorching the crown, lethally heating the cambium, or lethally heating the roots. The canopy, sub-canopy, shrub, and herb layers may be partially or completely killed depending on fire severity [24]. Vegetation recovery can begin within days after a fire and can take from a few years to many decades for the complete recovery to occur, depending on weather and climatic conditions.

The main causes of wildfire are lightning, volcanoes, and human action, the latter being the main cause in, for example, the densely populated Mediterranean Basin. The year 2017 was particularly significant in terms of the number of hectares burned in Southern Europe, especially in Italy. In August 2017, a wildfire covered the SW slope of Montagna del Morrone, involving an area of about 2200 ha. Geomorphological features of the study area were widely altered by the wildfire, increasing the possibility of triggering potential landslide as confirmed in August 2018, by the occurrence of a debris flow, which was triggered by an intense, short duration rainfall event. Debris flows are often the most common and most hazardous type of mass movement that can occur above a gully or steep couloir after a wildfire.

The Montagna del Morrone is located between the Maiella Massif to the NE and the Sulmona Basin to the west, inside the chain area of the Abruzzo Region (Figure 1). It is framed in the geological and geomorphological context of the Central Apennines and is characterized by a landscape that evolved as a result of the continuous combination of tectonics and selective erosion, which was strongly influenced by the juxtaposition of different lithological sequences due to tectonic activity (compressive, strike-slip, and extensional tectonics) and regional uplift [25–27].

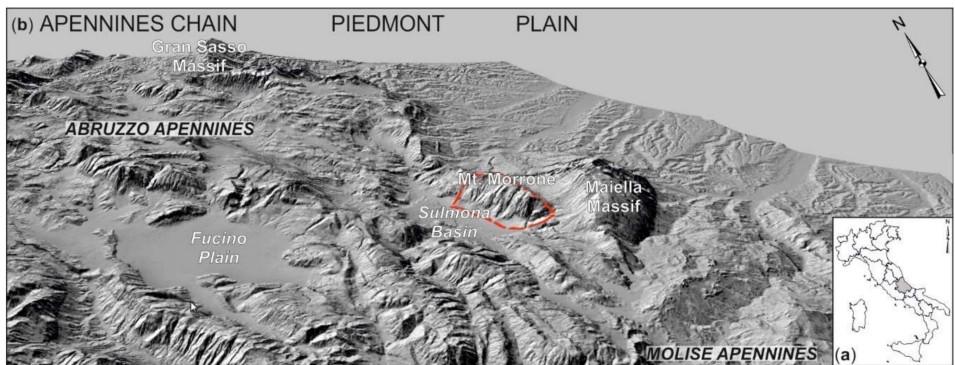

**Figure 1.** (**a**) Location map of the study area in Central Italy; (**b**) three-dimensional view (from 20 m DEM, SINAnet) of the Abruzzo Region. The red dashed line indicates the location of the study area.

The aim of this work was to geomorphologically analyse the SW slope of the Montagna del Morrone in order to assess the post-wildfire landslide hazard and to portray its spatial distribution in thematic maps.

## 2. Study Area

### 2.1. Geological and Geomorphological Setting

The Montagna del Morrone is one of the most eastern ridges of the Central Apennines chain and is located in the central-eastern sector of the Abruzzo Region. The relief features of the Central Apennines are represented by carbonate ridges (up to 2900 m a.s.l. high, i.e., Gran Sasso Massif, 2912 m a.s.l.; Maiella Massif, 2793 m a.s.l.) separated by longitudinal and transversal valleys, and wide intermontane basins (i.e., Fucino Plain, Sulmona Basin). The chain abruptly drops down to the

piedmont area (ranging from ~600 m a.s.l. to the coast line), which features a gentle cuesta and mesa landscape incised by SW–NE cataclinal valleys [25,26,28] (Figure 1).

The chain is made up of pre-orogenic carbonate lithological sequences pertaining to different Mesozoic–Cenozoic structural and palaeogeographic domains. Carbonate platform, slope, and pelagic basin sequences represent the carbonate backbone of the main ridges of the Abruzzo Apennines (Montagna del Morrone ridge, Maiella Massif), and pelagic sequences are widespread in the Molise Apennines featuring a chaotic assemblage on clayey-marly-limestone units. Sin-and late-orogenic deposits are represented by turbiditic foredeep sequences, which are largely covered and unconformably overlain by Pleistocene hemipelagic sequences with clay-sand-conglomerate marine transitional sequences. Post-orogenic deposits mainly consist of slope, fluvial and alluvial, and karst deposits. The main tectonic features are represented by NW–SE to N–S-oriented thrusts, which affected the chain from the Late Miocene to the Early Pliocene [25,29]. This compressional tectonics was followed by strike-slip tectonics along mostly NW–SE to NNW–SSE-oriented faults that were poorly constrained in age and largely masked by later extensional tectonic events since the Early Pleistocene [27,28,30] (Figure 2).

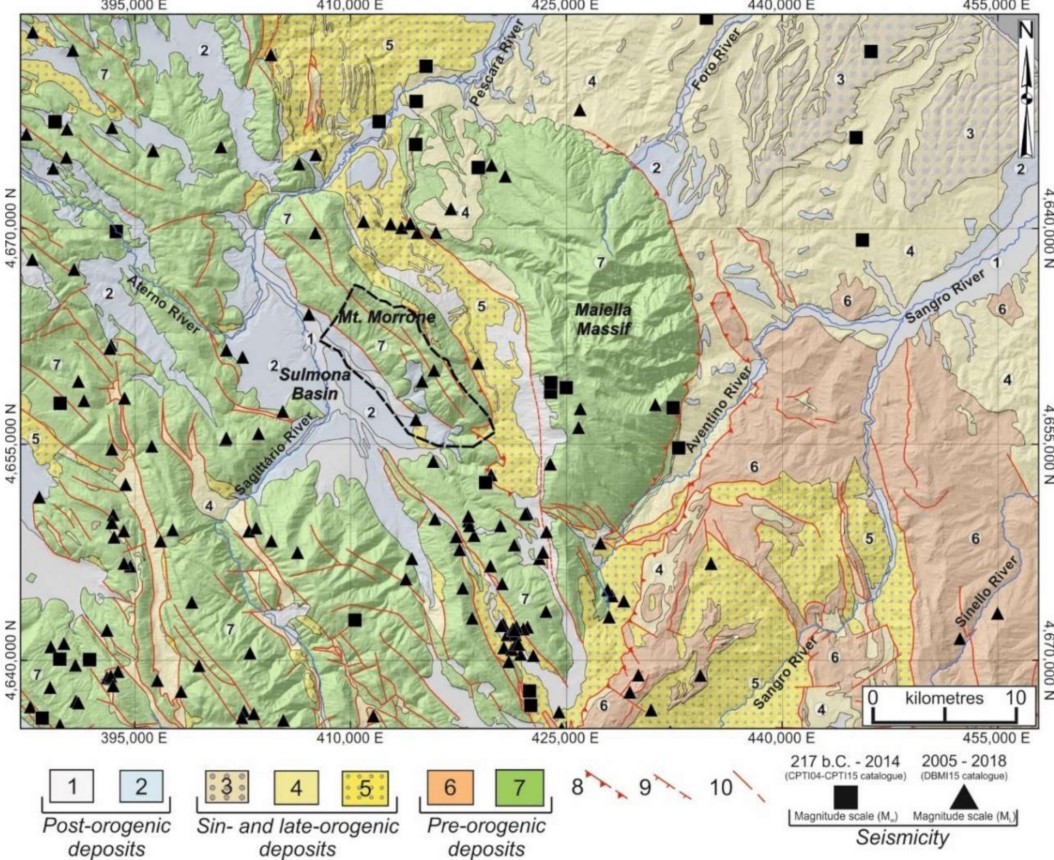

**Figure 2.** Geological scheme of central-eastern Abruzzo (modified from D'Alessandro et al. [31]) and the location of the study area (black dashed line). Legend: post-orogenic deposits—(1) fluvial deposits (Holocene), (2) fluvial and alluvial fan deposits (Middle-Late Pleistocene); sin- and late-orogenic deposits—(3) marine to continental transitional sequences (Early Pleistocene), (4) hemipelagic sequences with conglomerate levels (Late Pliocene–Early Pleistocene), (5) turbiditic foredeep sequences (Late Miocene–Early Pliocene); pre-orogenic deposits—(6) pelagic sequences (Oligocene–Miocene), (7) carbonate platform, slope and pelagic basin sequences (Jurassic–Miocene); (8) major thrust (dashed if buried); (9) major normal fault (dashed if buried); (10) major fault with strike-slip or reverse component (dashed if buried); seismicity—CPTI04 [32]-CPTI15 [33] catalogue (black square); DBMI15 [34] catalogue (black triangle).

The present day tectonic setting is characterized by extensional tectonics that is still active in the axial part of the chain, which is characterized by intense seismicity and strong historical earthquakes (up to M 7.0; e.g., Sulmona, 1706; Fucino 1915; Majella Massif, 1933; L'Aquila 2009) [35,36].

In this framework, the geological and geomorphological setting of the Montagna del Morrone ridge is the result of a complex cyclic evolution that occurred in succeeding stages with the dominance either of morphostructural factors, linked to the conflicting tectonic activity and regional uplift, or morphosculptural factors, linked to drainage network linear down-cutting and slope gravity processes. The slope architecture is, therefore, closely associated with the morphotectonic evolution, involving the entire ridge and the Sulmona Basin, which is due to the activity of the normal faults, which had uplift rates that strongly exceeded denudation rates between the Early and Middle Pleistocene [37,38].

The Montagna del Morrone is a ridge of considerable length (~20 km) and presents an articulated morphology, in both the longitudinal and transversal directions. The ridge is clearly lengthened in the NW–SE direction with markedly steep SW and NE slopes. It is made up of several peaks, the highest being Mt. Morrone (2061 m a.s.l.) in the middle of the group. Other peaks are, from NW to SE, Schiena d'Asino (1498 m a.s.l.), C.le Affogato (1783 m a.s.l.), C.le della Croce (1901 m a.s.l.), Mt. Cimerone (1849 m a.s.l.), and Mt. Mileto (1920 m a.s.l.) (Figure 3).

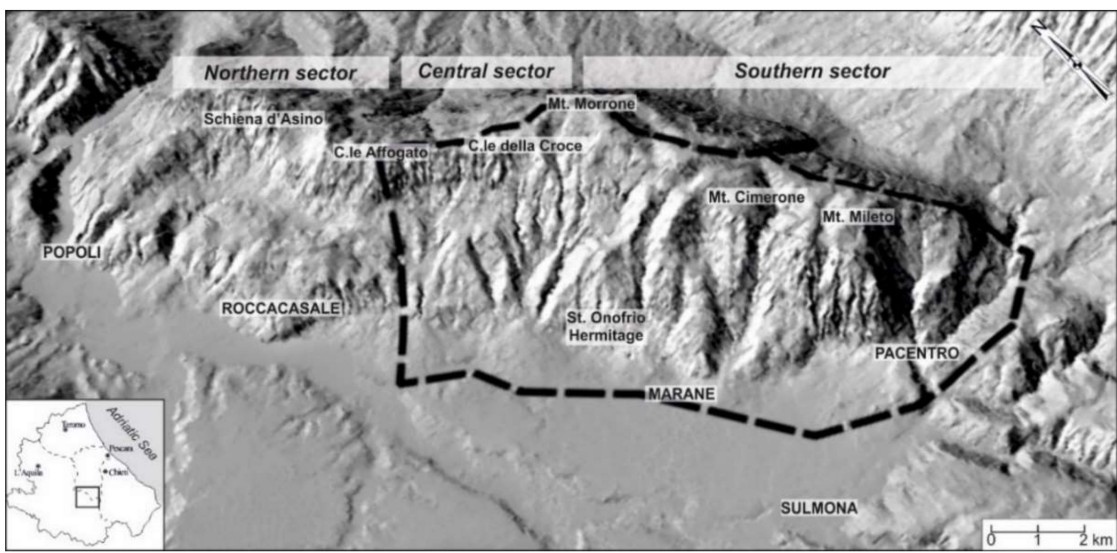

**Figure 3.** Shaded relief image of the Montagna del Morrone ridge. The black dashed box indicates the location of the study area.

On the eastern side, it lies next to an N–S-striking valley, which separates the ridge from the Maiella Massif. On the western side, it is bounded downslope by a sharp junction with the Sulmona Basin, which is crossed by several rivers (i.e., the Sagittario River and Aterno River) flowing into the Pescara River. In detail, the study area (black dashed box in Figure 3) is located in the central-southern sector of the SW slope, between C.le Affogato and Pacentro village.

The bedrock geology consists of carbonate successions which are represented by bedded carbonate rocks (C.le Affogato) made up of crystalline limestone, micritic limestone, and, in minor amounts, dolomite rocks in fine to medium thickness beds, referable to the slope-basin domain; massive carbonate rocks (from C.le Affogato to Mt. Morrone) made up of detritic limestone and calcarenites in a massive setting or locally bedded, referable to the margin domain; and carbonate rocks (from Mt. Morrone to Pacentro village) made up of compact limestone and calcarenites in thick beds, referable to the shelf domain [39].

The Quaternary continental deposits are present in the wide Sulmona Basin and along the slopes of the surrounding ridges. They can be referred to paleo-landslide deposits, talus slope and debris cones, fluvial deposits, alluvial fans deposits, and eluvial and colluvial deposits [38–43].

From a tectonic standpoint, the SW slope is cut by two main fault systems in the NW–SE direction, dipping SW. One fault line is located at the base of the slope, corresponding to the junction with the Sulmona Basin, while the other is located in the mid-slope [36–38,43–45].

### 2.2. Montagna del Morrone Fire—August 2017

The 2017 Montagna del Morrone fire occurred along the central-southern sector of the SW side of the ridge, in the area between C.le Affogato and Pacentro village. Previously, according to the official data of the Majella National Park (time period between 1997 and 2012) [46], no major fire events have been recorded in the study area.

In detail, the 2017 wildfire occurred over a total duration of 24 days and burned about 2200 ha (Figure 4a). The fire developed with varying severity levels depending on the vegetation cover and lithological characteristics on 19 August and primarily burned the area near the Pacentro village. Subsequently, a second trigger point was identified above the Marane village. Overall, the fire developed rapidly along the entire slope and then slowly descended toward the junction with the Sulmona Basin, opening new fronts downwards without finding major obstacles.

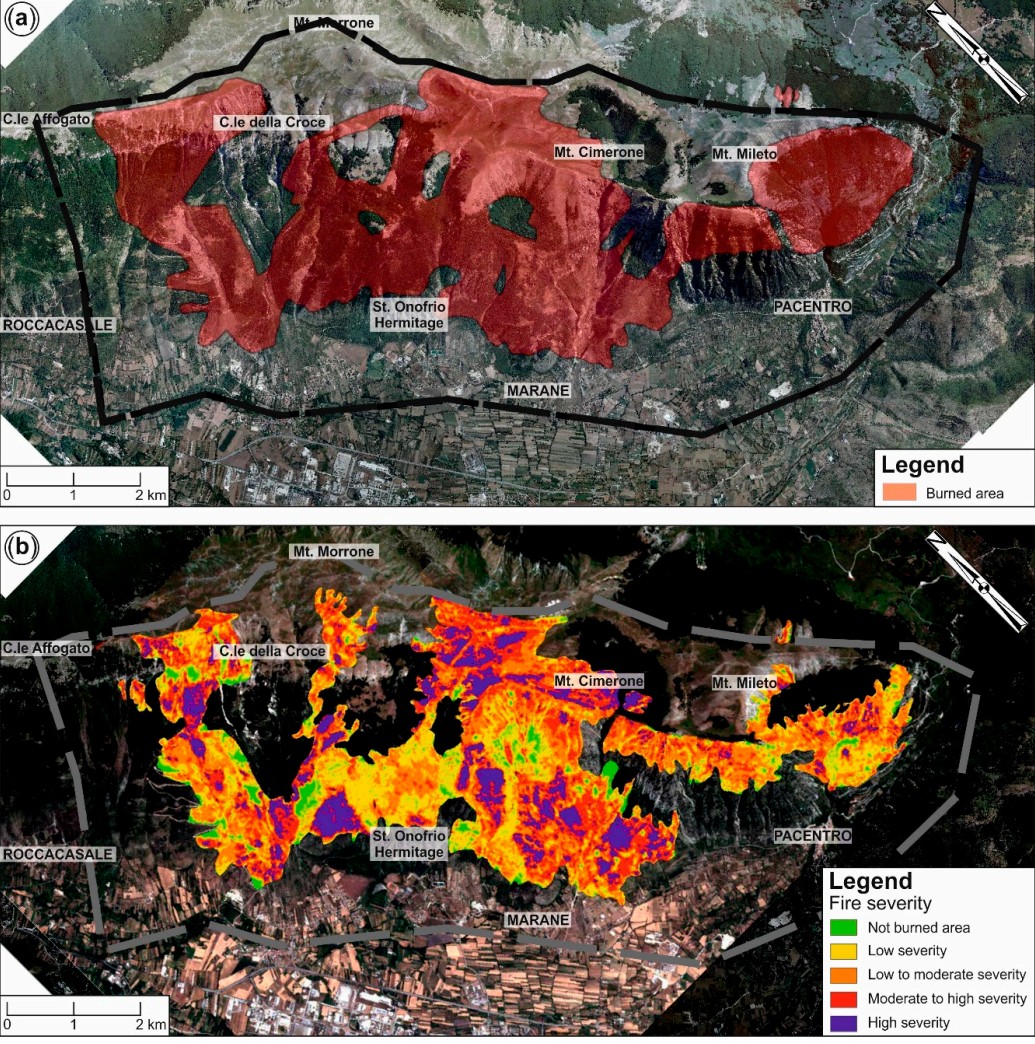

**Figure 4.** (**a**) August 2017 fire perimeter map; (**b**) fire severity map (modified from Frate et al. [47]). The dashed box indicates the location of the study area.

Fire severity was taken into account analyzing the map from Frate et al. [47] (Figure 4b); the authors defined the severity classes using pre and post Sentinel 2-A data of the burned areas, from which the normalized burned ratio index [48] was obtained. The classification results as follows: Not-burned area (5%, 112 ha), low severity (23%, 510 ha), low to moderate severity (33%, 722 ha), and moderate to high severity (26%, 574 ha). The high severity class covered 13% (267 ha) of the study area and was concentrated especially on the upper-slope near Mt. Cimerone, on the mid-slope between the alignment of C.le Affogato–C.le della Croce, and on the toe-and mid-slope above the Marane village and near St. Onofrio Hermitage.

### 2.3. Vegetation Cover

The pre-wildfire vegetation cover was rich, complex, and variously articulated as a consequence of the local climatic conditions, the considerable altitude extension, and the physiographic features as well as the presence of man. According to forest-typological maps of Abruzzo Region [49], the pre-wildfire vegetation cover was classified into different categories (Figure 5), as follows:

- Meadows, widespread near the crest line between 1700–2000 m a.s.l.;
- Beeches, located next to C.le Affogato and Mt. Mileto (above 1800 m a.s.l.);
- Conifers, along the slope from Roccacasale to Marane village, at altitudes between 500 and 1700 m a.s.l.;
- Shrublands, downstream of Mt. Cimerone at an altitude between 500 and 700 m a.s.l.;
- Absent, in the urban area at the basal-slope and in correspondence of rock outcrops.

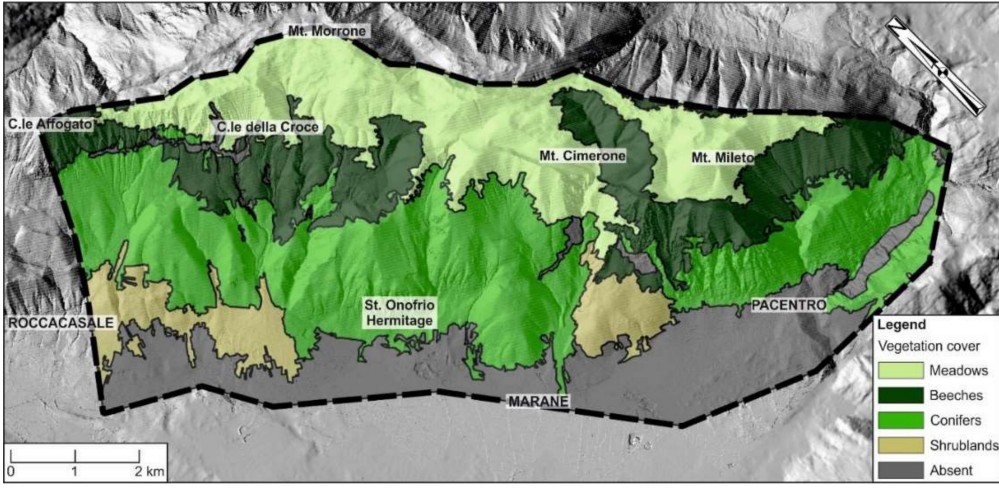

**Figure 5.** Pre-wildfire vegetation cover map (modified from [49]). The black dashed box indicates the location of the study area.

The vegetation features were widely changed by the 2017 fire. In order to evaluate the post-wildfire landslide hazard, it was reclassified, according to its size, appearance, and distribution in areas with meadow, arboreal, and shrubby or absent vegetation. Zones affected by the wildfire were classified as absent vegetation cover areas. In particular, the meadow areas are widespread in the upper-slope, the arboreal ones are present especially in the mid-slope, while the shrubby or absent vegetation areas are distributed uniformly among all slopes, highlighting the areas affected by the wildfire (Figure 6).

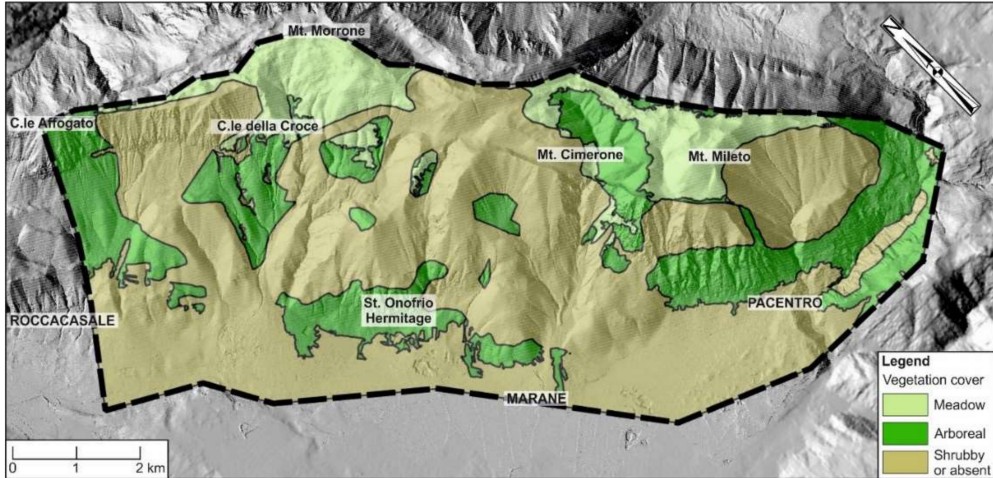

**Figure 6.** Post-wildfire vegetation cover map. The black dashed box indicates the location of the study area.

## 3. Methods

The SW slope of the Montagna del Morrone was investigated through a detailed geomorphological analysis incorporating (i) morphometric and slope analysis, (ii) rainfall data analysis, (iii) geological and geomorphological field mapping, (iv) geomechanical investigations, and (v) landslide hazard assessment.

The analysis was performed using topographic maps (1:25,000–1:5000 scale) and supported by a digital terrain model (1 m LIDA—Laser Imaging Detection and Ranging).

Morphometric and slope analysis was carried out with the GIS software (QGIS 2016, version 2.18 "Las Palmas" and ArcMap® 10.1, ESRI, Redlands, CA, USA). It was based on the definition of the drainage network and the orographic features of the study area. Basin boundaries and drainage lines were digitized from 1:5000 topographic maps and verified by means of 1:33,000 and 1:10,000 air-photos. The study area was classified into 17 basins, that include at least a second order stream, and four areas grouped into "hydrographic units", which show a homogeneous distribution of first order stream, gullies and channels along steep to sub-vertical scarps. This analysis led to define a total of 21 basins, of which basic morphometric parameters (such as area, perimeter, relief, length) were obtained from the DEM. These parameters were applied to obtain the Melton ratio (R) [50,51], expressed by the equation:

$$R = (H_{max} - H_{min})/A^{0.5} \tag{1}$$

where $H_{max}$ and $H_{min}$ are the maximum and the minimum relief, respectively, and A is the basin area.

Several studies [51–53] have demonstrated that the Melton ratio in addition to the basin length can be used to differentiate between basins prone to debris flows and debris floods. The combination of this appropriate pair of attributes identifies the class boundaries of basins prone to debris flows with Melton ratio values >0.6 and basin length values <2.7.

The morphometric analysis was based on two main parameters: Slope and energy of relief. The slope was calculated as the first derivate of elevation [54]. The local relief was calculated as the elevation range within $1 \times 1$ km windows, in accordance with Ahnert [55].

The rainfall dataset was obtained from the analysis of a network of 12 pluviometric stations. Using the ArcGIS Kernel Interpolation function, the rainfall density map of the study area was prepared, considering a 30-year time record (1987–2017). For the specific August 2018 event, the data enabled the analysis and comparison of hourly rainfall intensity, event cumulative rainfall, event duration, and daily rainfall.

Geological and geomorphological analyses were based on field mapping and stereoscopy and air-photo interpretation. Field mapping was carried out at the 1:5000 scale to investigate lithological features, superficial deposit cover, and the type and distribution of geomorphological landforms (structural, slope, and fluvial). The mapping was performed according to the guidelines of the Geological Survey of Italy and AIGeo [56–59] and was in accordance with the literature concerning geomorphological mapping [60–63]. Air-photo interpretation was performed using 1:33,000 and 1:10,000 scale stereoscopic air-photos (Flight GAI 1954 and Flight Abruzzo Region 1981–1987) and 1:5000 scale orthophoto color images (Abruzzo Region 2010). Combined with the field mapping, this supported the geomorphological investigation of the study area. Finally, a drone survey was performed to support the geomorphological analysis of the August 2018 debris flow event and to evaluate its contributing area.

Geomechanical investigation was carried out to analyze the scan line survey data gathered from four sites. The susceptibility of the slopes to the rockfall was analyzed by investigating the geomechanical features of the slopes [64] and outlining possible movement mechanisms and block dimensions. The spacing and persistence of discontinuity sets were analyzed together with their relationship with the strata attitude and slope orientation in order to define the size of blocks susceptible to fall as well as their fall mechanism [10]. Kinematic analysis was performed using the modeling software (Rocscience Inc. Dips 2018, version 7.012) based on Markland's test. According to Markland [65] and Hoek & Bray [66], the direction of failure is linked to the intersection of the slope orientation, rock discontinuities, and strata attitude, considering also the internal friction angle of the rock mass.

The post-wildfire landslide hazard analysis was performed following a heuristic or expert-based approach, which was developed and supplemented using GIS technology [15,17,67–70]. In this selected method, the type and degree of hazard is derived from the geomorphological expert judgment, based on either direct (field mapping) or indirect (GIS data processing) analysis [22]. This approach led to the identification of relevant parameters for the mechanism of landslide occurrence in a mountainous forest context. Regarding the study area, five factors (slope, lithological features, bedrock fracturing, and geomorphological elements, post-wildfire vegetation cover) were considered as instability factors and portrayed in thematic maps. In this study, each attribute was rated between 1 and 10 considering the importance of the five factors in the landslides occurrence, and according to its criteria and priority, defined by expert opinion [71,72]. The analysis of interactions between the attributes' weights allowed the achievement of a geomorphology-based matrix, which defined four classes of landslide hazard (low, moderate, high, and very high). Finally, all the collected data were integrated into the GIS software through a cartographic and weighted overlay process [72–75] in order to portray the spatial distribution of the landslide hazard.

## 4. Results

### 4.1. Orography and Hydrography

The study area can be subdivided into three different slope units according to the orography of the landscape (elevation, slope, energy of relief) (Figure 7). The toe-slope unit is highlighted by the junction area with the Sulmona Basin; it is characterized by the flat morphology (<300 m a.s.l.) of the eastern portion of the basin passing through a range of altitude between 300 and 600 m a.s.l. It shows the lowest slope values (between 0% and 15%), and the energy of relief ranges from 5 to 110 m. The mid-slope unit features a varied morphology and is characterized by several wide scarps and valleys; the elevation rises from 600 to 1600 m a.s.l. with slope values ranging from 20% to 80%, located especially in the southernmost sector of the ridge, with sub-vertical scarps showing the highest slope values (>500%) downstream of the Mt. Mileto escarpment and St. Onofrio Hermitage. The energy of relief is distributed along this slope unit: Low values ranging from 300 to 600 m are located especially in the northern sector, while medium values (from 600 to 900 m) are present along the steep

scarps. The highest values of the energy of relief are located downstream of Mt. Mileto escarpment ranging 940 m. The upper-slope unit is characterized by the maximum elevation (from 1600 to >2000 m a.s.l.) and includes the highest peaks of the ridge. The northernmost sector is highlighted by slope values of 500% and energy of relief from 800 to 940 m along the sub-vertical scarps near the ridge of C.le della Croce–C.le Affogato. The remaining portion of the upper-slope is characterized by slope values from 5 to 50% and energy of relief values from 400 to 600 m, heterogeneously distributed in an area of slight incline.

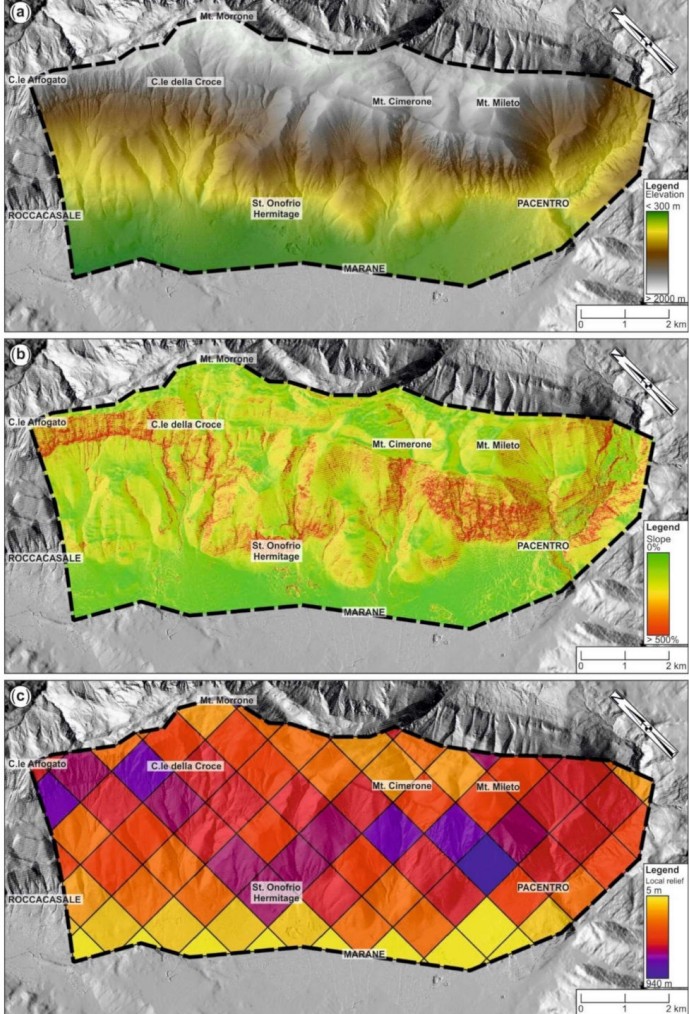

**Figure 7.** Physiographic features of the study area: (**a**) elevation map; (**b**) slope map; (**c**) local relief map. The dashed black line shows the study area.

The hydrography of the study area is characterized by minor and ephemeral stream channels, which flow in the NE–SW direction. The drainage network shows a mainly sub-parallel drainage pattern.

The SW slope is subdivided into 21 basins (1–21) (Figure 8). Of these 21 basins, seven are spread from the crest line to the base of the slope (1,4,5,9,12,14, and 21). Four are endorheic (3,8,13, and 19) and are located on the upper-slope. Five basins (6,11,15,17, and 18) have developed on the mid-slope with a closure in the toe-slope. Four of these (2,7,10, and 20) developed across the toe- and mid-slope and can be considered hydrographic units, characterized by a homogeneous distribution of gullies and channels along steep to sub-vertical scarps. The relief of these basins varies from a minimum of 266 m at endorheic basin 9 to a maximum of 1510 m at basin 4, which extends down from the highest peak (2061 m a.s.l., Mt. Morrone) right to the toe-slope.

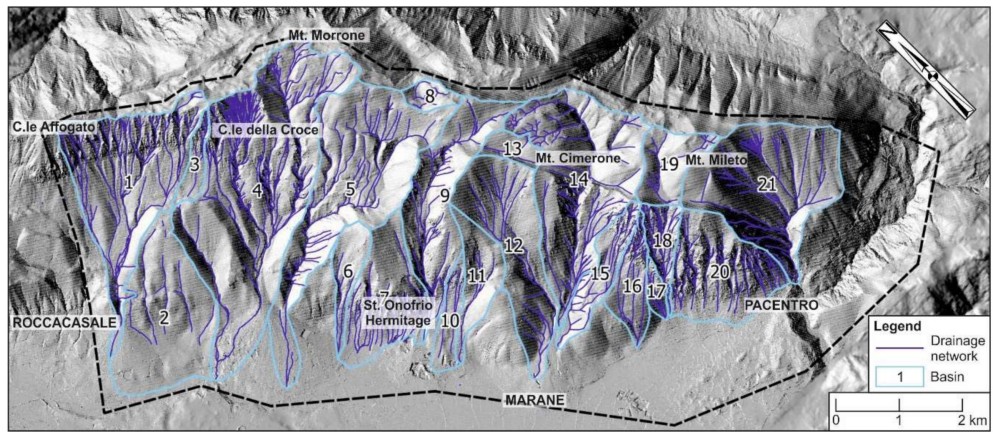

**Figure 8.** Drainage network and basin subdivisions. The dashed black line shows the study area.

The defined basins (Figure 8) show a complex organization and a variable geometry, from elongated to squared off or sub-circular. In the northern-central sector, the endorheic basins (3 and 8) are located near the principal crest line, while the remaining basins have developed along the slope and present an elongated shape with a clear narrowing downstream (basins 1,4,5, and 9). The hydrographic network shows a parallel pattern with some sub-dendritic sections in the upper-slope near the alignment of C.le della Croce–C.le Affogato. In the southern sector, three basins (12,14, and 21) have developed from the upper- to the toe-slope with an elongated shape towards the downstream area, while, the remaining basins (10,11,15,16,17,18, and 20) have developed from the mid-slope with the closure in the toe-slope. An isolated endorheic basin (19) in the upper-slope features a rectangular shape and a relatively rectangular drainage pattern.

The main morphometric characteristics of the basins delineated in the study area are summarized in Table 1 and graphically shown in Figure 8.

**Table 1.** Main morphometric parameters of the 21 basins in the study area.

| Basin | Area (km²) | Perimeter (km) | Max. Relief (m a.s.l.) | Min. Relief (m a.s.l.) | Relief (m) | Melton Ratio | Length (km) |
|---|---|---|---|---|---|---|---|
| 1 | 3.900 | 11.541 | 1886 | 374 | 1511 | 0.766 | 4.458 |
| 2 | 3.164 | 8.101 | 1060 | 307 | 753 | 0.423 | 1.735 |
| 3 | 0.377 | 3.793 | 1860 | 1053 | 806 | 1.314 | 2.803 |
| 4 | 7.277 | 14.196 | 2059 | 323 | 1736 | 0.644 | 5.581 |
| 5 | 4.651 | 13.600 | 1995 | 356 | 1639 | 0.760 | 5.028 |
| 6 | 0.634 | 4.758 | 1376 | 365 | 1010 | 1.270 | 2.210 |
| 7 | 2.003 | 6.887 | 1431 | 351 | 1080 | 0.763 | 2.655 |
| 8 | 0.311 | 2.415 | 2000 | 1825 | 174 | 0.314 | 0.529 |
| 9 | 2.465 | 10.166 | 1998 | 431 | 1567 | 0.998 | 4.155 |
| 10 | 0.533 | 4.369 | 1225 | 373 | 851 | 1.167 | 1.986 |
| 11 | 0.947 | 5.818 | 1578 | 385 | 1192 | 1.226 | 2.538 |
| 12 | 2.644 | 8.733 | 1846 | 424 | 1422 | 0.875 | 3.613 |
| 13 | 0.369 | 2.943 | 1885 | 1720 | 165 | 0.272 | 0.522 |
| 14 | 4.253 | 10.770 | 1884 | 563 | 1321 | 0.641 | 3.794 |
| 15 | 0.912 | 5.840 | 1799 | 512 | 1286 | 1.348 | 2.623 |
| 16 | 0.785 | 4.803 | 1755 | 446 | 1309 | 1.477 | 2.181 |
| 17 | 0.211 | 2.338 | 1207 | 526 | 680 | 1.483 | 1.058 |
| 18 | 0.578 | 4.476 | 1799 | 532 | 1266 | 1.667 | 1.854 |
| 19 | 0.952 | 4.575 | 1919 | 1653 | 266 | 0.273 | 1.466 |
| 20 | 2.294 | 6.223 | 1731 | 524 | 1207 | 0.797 | 2.035 |
| 21 | 3.787 | 8.520 | 1905 | 654 | 1251 | 0.643 | 2.648 |

*4.2. Rainfall Event*

The analysis of the precipitation data, which was evaluated through the analysis of 12 different pluviometric stations from a 30-year time record (1987–2017), proved certain features of the climate in

the region (Figure 9). The study area features a Mediterranean to warm temperate mountainous climate type, with a general bimodal pattern marked by an absolute maximum in autumn and a relative one in spring, and an absolute minimum in summer [76,77].

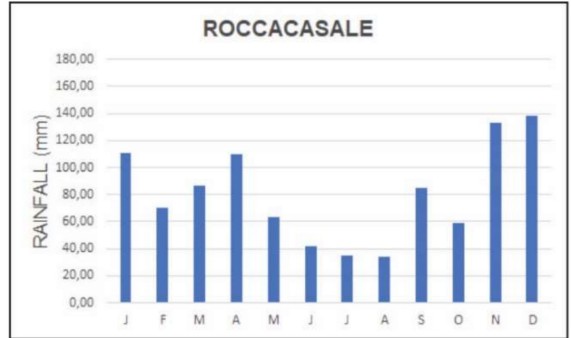
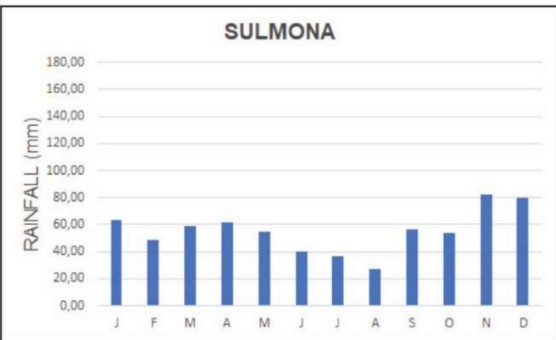

**Figure 9.** Pluviometric regime of Roccacasale and Sulmona pluviometric stations, located next to the study area.

The average annual precipitation is 700–1400 mm/year, with occasional heavy rainfall (>100 mm/day and 30–40 mm/h) [78]. This arrangement is graphically shown in Figure 10 with the lowest values located at the Sulmona station and the highest one located at Passo Lanciano station.

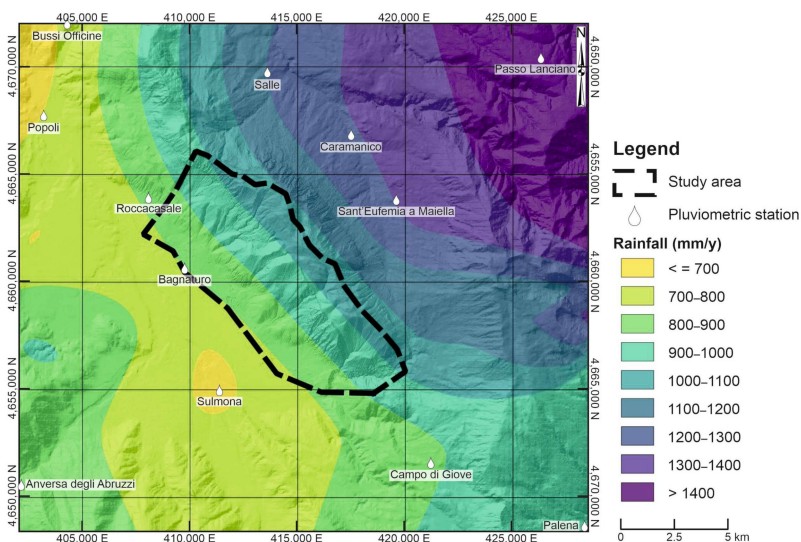

**Figure 10.** Average annual rainfall map. White dots represent the pluviometric stations.

Heavy Rainfall Event of August 2018

In August 2018 a monthly rainfall of about 150 mm (Figure 11a) was recorded at the Sulmona station, showing a significant difference in the pluviometric regime compared to the average monthly rainfall, recorded (Figure 9). In this framework, an intense, short duration rainfall event occurred on 15–16 August and was characterized by ~95 mm of rainfall in 48 h; in particular, the cumulative rainfall was about 90 mm in 12 h, from 20:00 on 15 August to 08:00 on 16 August. The intensity was moderate, with low values ranging from 3 to 11 mm/h and some peaks ranging from 18 to 22 mm/h (Figure 11b).

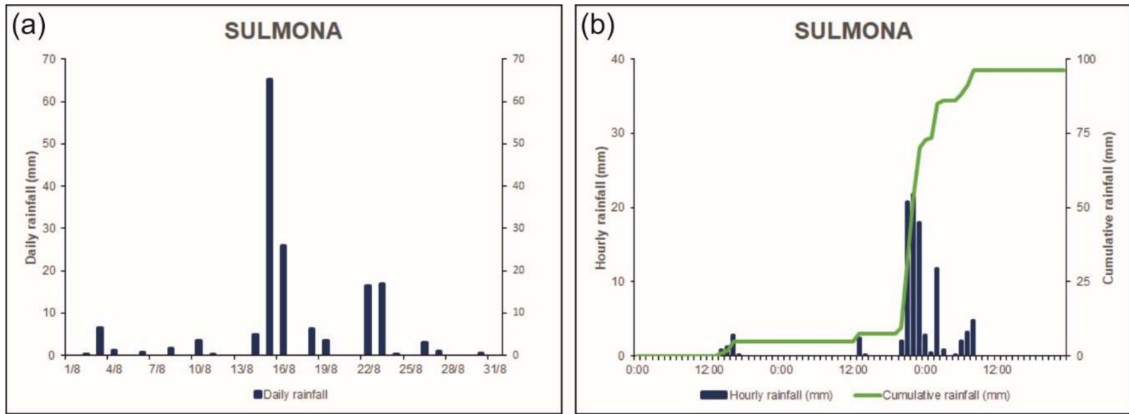

**Figure 11.** Sulmona pluviometric station: (**a**) Daily rainfall in the month of August 2018; (**b**) hourly and cumulative rainfall during the event of 15–16 August 2018.

*4.3. Lithological Data*

Lithological data were classified according to their nature, the degree of cementation and fracturing, and their behavior towards degradation processes. Bedrock lithology was modified from Miccadei et al. [38] according to the post-wildfire setting of the study area. In detail, bedrock fracturing, due to the intense tectonic activity of the study area, combined with the fire activity, increased landslide instability. The geological setting is graphically summarized in the geolithological map of Figure 12.

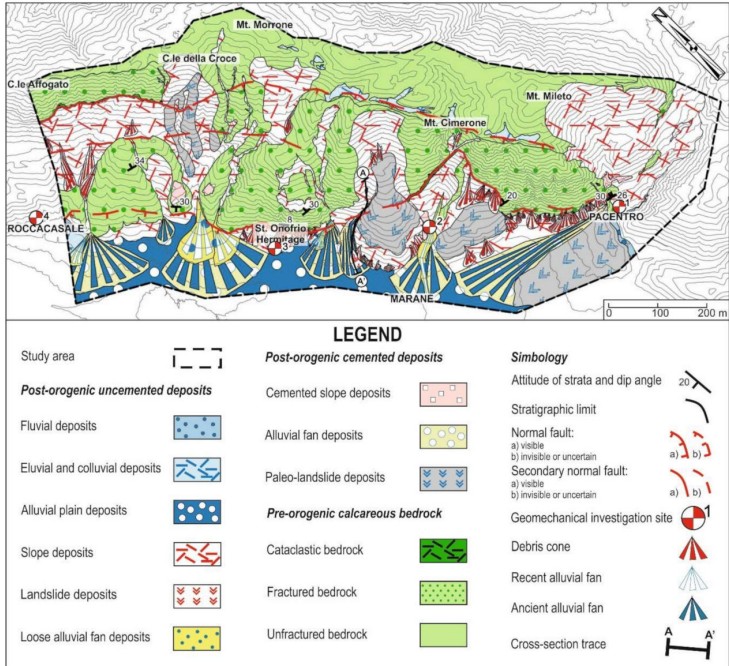

**Figure 12.** Geolithological map of the study area. The cross-section trace matches with the August 2018 debris flow channel.

Lithological data were classified as:

- Pre-orogenic calcareous bedrock;
- Post-orogenic cemented deposits;
- Post-orogenic uncemented deposits.

### 4.3.1. Pre-Orogenic Calcareous Bedrock

These lithologies were distinguished according to the degree of fracturing as follows: Unfractured bedrock, which is composed of alternating dolomitic limestone and micritic limestone present in the upper-slope, and is characterized by a moderate to low fracturing rate; fractured bedrock, which is composed of micritic limestone, biodetritic limestones, and dolomitic limestone present in the mid- and toe-slope and is characterized by a moderate to high fracturing rate; and cataclasite, which is composed of limestone in gray-whitish metric levels, is well recognizable in the mid-slope, and is characterized by a high fracturing rate.

### 4.3.2. Post-Orogenic Cemented Deposits

These deposits are made up of clastic lithologies due to paleo-landslides, alluvial fans, and cemented slope deposits. In detail, paleo-landslide deposits consist of heterometric chaotic breccias with limestone elements that are always angular with blocks up to 2–3 m in diameter and mud-supported with a whitish silt-clay matrix; alluvial fan deposits consist of conglomerates and breccias with heterometric calcareous clasts, containing elements of up to 2–3 m in diameter; cemented slope deposits consist of mainly calcareous breccias and conglomerates with a heteromeric nature and clasts with decimetric to metric dimensions, arranged in flat-parallel strata or with uncertain internal stratification. They are well-cemented deposits with a whitish to pinkish matrix.

### 4.3.3. Post-Orogenic Uncemented Deposits

They can be distinguished in: Loose alluvial fan deposits, which are composed of gravel and heteromeric and poorly consistent conglomerates with sub-rounded calcareous clasts. These deposits are usually entrenched in the oldest post-orogenic deposits or in the calcareous bedrock; landslide deposits, which are made up of loose limestone and chaotic material. They are distributed mainly in the toe-slope and subordinately in the mid-slope; slope deposits, which consist primarily of loose heterometric calcareous material, located at the base of the slope and downstream of the several couloir and/or gullies; alluvial plain deposits, which consist of alternating brown sands and mainly calcareous conglomerates with well-rounded pebbles from centimetric to decimetric size inside a silt-sandy matrix; eluvial and colluvial deposits, made up of sands, silts, and gravels containing material due to the re-elaboration of the oldest deposits; and fluvial deposits, made up of sands, gravels, and limes with centimetric pebbles.

### *4.4. Tectonic Features*

There are two main normal fault systems with a predominantly N40–50W orientation and SW dipping. These systems displace the pre-orogenic calcareous bedrock and the post-orogenic deposits and are clearly highlighted by morphological evidence at different heights on the slope corresponding to sharp slope breaks or clear fault scarps.

From the base upwards, the main fault systems are as follows: Basal border fault, a system of normal faults with an attitude of N50–60E, 50SW placed at heights between 750 and 800 m a.s.l. (St. Onofrio Hermitage, Pacentro); Schiena d'Asino fault, made up of a system of normal faults with a N20–30W strike and 60°–50° SW dip located at heights between 1100 and 1400 m a.s.l. near the alignment of C.le Affogato–C.le della Croce. These fault planes, in the northern part, are characterized by large rock fault scarps, while southwards, they show a clear reduction of displacement and morphological evidence.

Minor faults with a NE–SW orientation and limited extension are also present transverse to the slope, mostly in the northern sector, and they can be interpreted as transfer elements between the Schiena d'Asino fault and the basal border fault.

### 4.5. Geomechanical Investigations

Investigations were performed in order to verify the bedrock fracturing and to outline possible movement mechanisms and block dimensions. A detailed scan line survey was carried out at four different sites and a kinematic analysis was performed. The investigation sites were selected according to their geological and geomorphological features. The statistical analysis of discontinuity orientation and spacing and their relationship with slope orientation allowed us to define the main discontinuity sets and the fall mechanism.

Joint spacing and persistence, and discontinuity sets were analyzed together with their relationship with slope orientation in order to define the size of blocks susceptible to falling. In detail, the main discontinuity sets were oriented at N30W 60SW and N78E 70SE, and their spacing and persistence allowed us to infer the size of unstable blocks of large ($Jv = 1/3$ joint/m$^3$) to medium ($Jv = 3/10$ joint/m$^3$) sizes [64] (Table 2).

**Table 2.** Geomechanical investigation data: Slope orientation, strata attitude (S0), and main joints (J1,2,3,4) analyzed (for investigation sites' locations, see Figure 11); estimated Jv (joint/m$^3$) for each site.

| | Site | Slope | S0 | J1 | J2 | J3 | J4 | Jv |
|---|---|---|---|---|---|---|---|---|
| 1 | Pacentro | N88E 65SE | N81W 32NE | N08W 59SW | N02W 65E | N24W 59SW | N72E 80SE | 8 |
| 2 | Marane | N06W 75SW | N60W 21SW | N86E 65SE | N11E 70SE | - | - | 2.5 |
| 3 | St. Onofrio Hermitage | N80W 80SW | N40W 09SW | N12W 80SW | N33E 81SE | - | - | 9 |
| 4 | Roccacasale | N57E 77SE | N40W 53NE | N51W 64SW | N08E 63SE | N60E 65NW | N30E 68SE | 5 |

The results of the kinematic analysis show a prevailing toppling failure. This analysis features a variable trend with possible toppling directions SSW-oriented at site 1, W-oriented at site 2, SSW-oriented at site 3, and finally, SSE-oriented at site 4 (Figure 13).

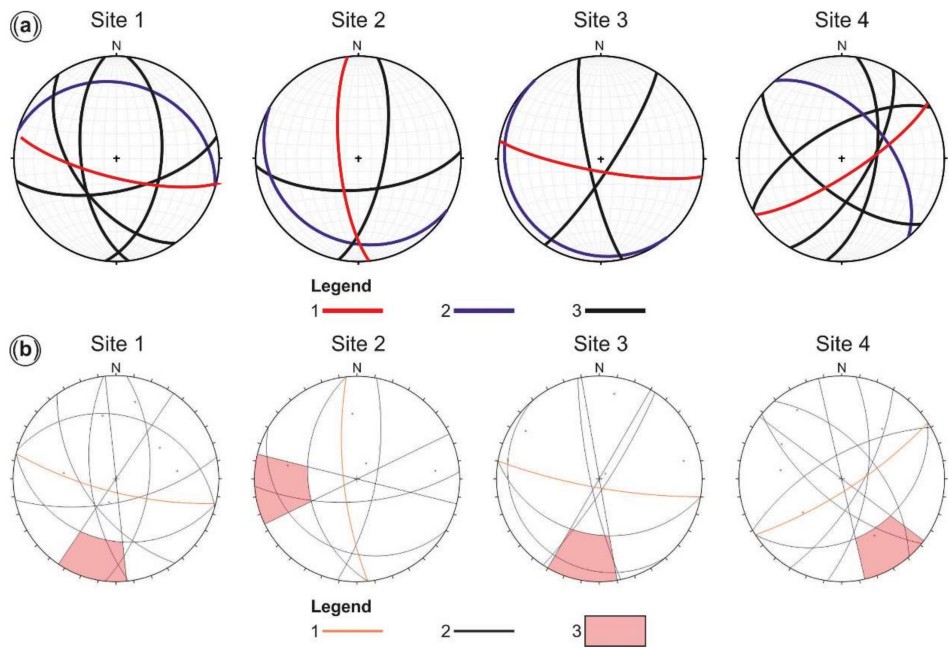

**Figure 13.** (**a**) Plots of the main discontinuity sets (strata and joints; lower hemisphere projection); for investigation sites' locations see Figure 11. Legend: (1) Slope orientation; (2) strata attitude (S0); (3) main joints (J1,2,3,4); (**b**) markland test plot. Legend: (1) Slope orientation; (2) discontinuity sets; (3) potential toppling zone.

### 4.6. Geomorphological Data

#### 4.6.1. Structural Landforms

Fault scarps are located at various heights along the slope, corresponding to the main fault system. They are represented by rock scarps from some tens of meters to 100 m high that are markedly straight. The scarps are made up of generally well-smoothed scarplets from a few decimeters to some meters high with slope values >100%. The retreated and weathered fault scarps can be identified as weak breaks in the slope that are often discontinuous and partially or completely covered by surface deposits. They develop particularly where the calcareous bedrock is made up of highly fractured limestone and cataclasite (along the basal fault line at heights of 500–1300 m), because these structural conditions favour weathering processes. The principal crest lines are located on the upper-slope with a NW–SE direction, while the secondary crest line is a lower and rounded one, located on the mid-slope (Figure 14).

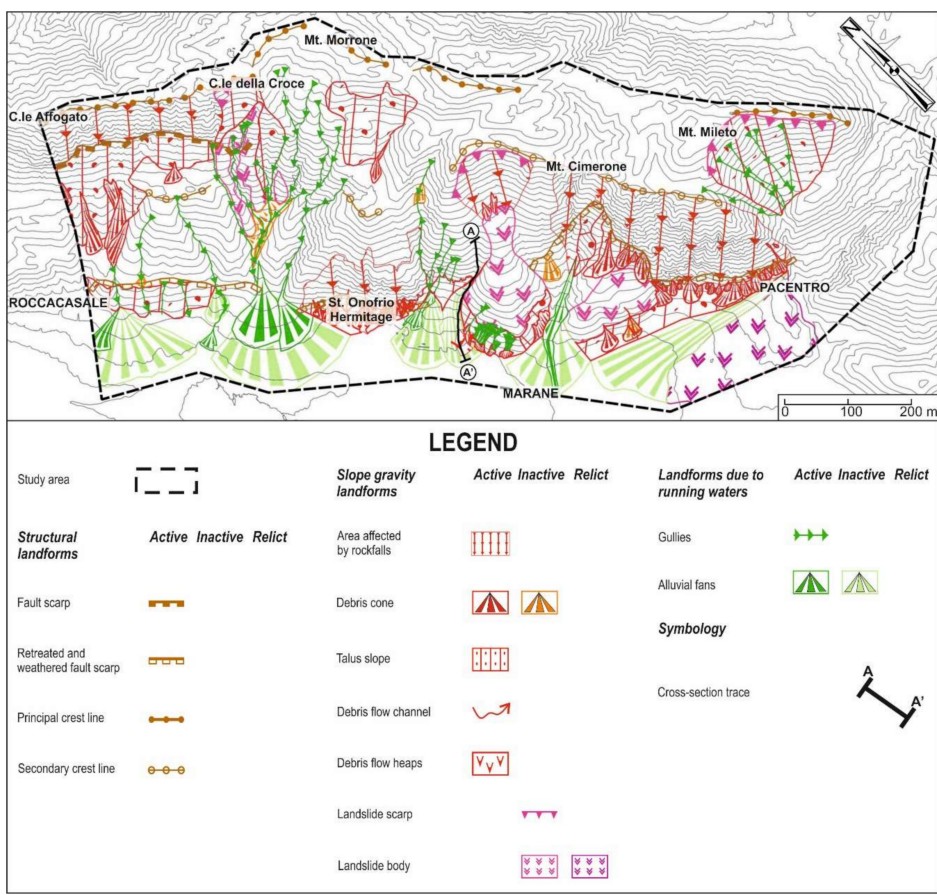

**Figure 14.** Geomorphological map of the study area. The cross-section trace matches with the August 2018 debris flow channel.

#### 4.6.2. Slope Gravity Landforms

Rock scarp affected by rockfalls are located in the alignment between C.le Affogato and C.le della Croce, near the St. Onofrio Hermitage and near the Pacentro village; they show a general NW–SE orientation with varying height ranging from 2–3 to few tens of meters. The couloirs with debris discharge are arranged perpendicularly to the slope and are widely developed in the study area; they are affected by polygenic processes such as debris discharge, water runoff, and snow avalanches. According to their spatial distribution along the slope, these landforms were grouped into a more general landform called area affected by rockfalls. Talus slope and debris cone are formed by bodies

of heterogeneous calcareous material with blocks of variable dimensions from centimetric to metric, which, in general, are arranged homogeneously and graded proceeding towards the base of the cone; they are present along the whole slope, especially near C.le della Croce at heights ranging from 1300 to 700 m a.s.l, near St. Onofrio Hermitage, at heights from 600 to 400 m a.s.l., and downstream of Mt. Mileto at heights from 600 to 800 m a.s.l. Debris flows are widely present in the area especially close to the toe-slope (near the St. Onorfrio Hermitage and Marane); they are characterized by a chaotic organization in the accumulation of the material with very low selection and the presence of clasts aligned at the base of the invasion area. The rockslide body is made up of bodies of limestone in large blocks and heterometric carbonate breccias in a chaotic arrangement with the abundant clay-silt matrix. They are distributed both at the toe-slope and scattered on the mid-slope. Landslide scarps are made up of arched or semi-circular rocks scarps that have generally been weathered and shaped by further slope gravity processes (Figure 14).

### 4.6.3. Landforms Due to The Running Waters

The gullies are widely present along the slope in the NE–SW direction. The alluvial fans are widespread in the junction area between the toe-slope and the plain; their apexes are located close to the basal border fault downstream of the main gullies near Pacentro and Marane village. These landforms are mostly inactive, but, at some points, there are some active fans, embedded one inside the other, with the apexes characterized by evidence of recent activity (Figure 14).

### 4.6.4. Debris Flow Event

On the evening of 15 August 2018, an intense, short duration rainfall event hit the study area and triggered a debris flow in the area of Casato Santa Lucia near Marane village, an area widely affected by the 2017 Montagna del Morrone fire.

A detailed field survey, integrated with a drone survey and mapping, was performed to better investigated the geomorphological features and to evaluate the debris flow' contributing area, defining homogenous areas on the couloir, classified as detachment, transit, and invasion area (Figure 15).

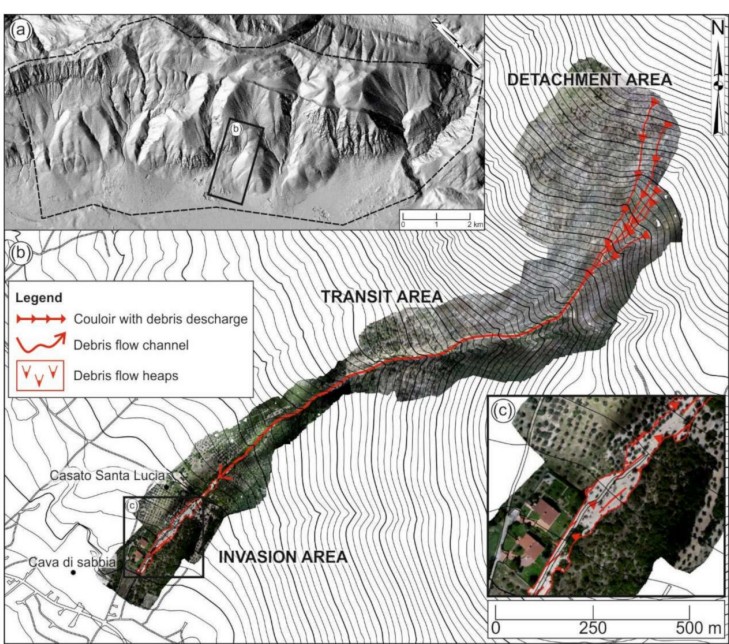

**Figure 15.** (**a**) Location map of the debris flow's area; (**b**) high resolution orthophoto (mesh 25 × 25 cm) showing the debris flow event of 16 August 2018; (**c**) zoomed view of the invasion area.

This survey revealed that no clear initiation failure could be observed, thus promoting the idea that the flow probably started by erosion along rills in the higher parts of the couloir and transformed into a mature debris flow in the subsequent transit area. Several secondary failures of adjacent couloirs were also observed, which enlarged the total volume of the debris flow. As the flow moved down, the solid material content increased due to scouring, shallow lateral slides, and the incorporation of burned trees. The main volume of the debris flow was deposited in the invasion area.

As revealed in Cannon & Gartner [79], debris flows following wildfire are often triggered by the first heavy rainstorm occurring in the burned area, as testified by the ~90 mm of precipitation recorded at the Sulmona station between 20:00 on 15 August and 08:00 on 16 August 2018.

The geological cross section shows the pre- and post-wildfire setting of the debris flow channel in the area of Casato Santa Lucia. In the pre-wildfire framework (Figure 16a), much of the sediment is stored upslope of vegetation cover, especially stems, shrubs, and herb layers. Following the destruction of vegetation by fire, as shown in the post-wildfire section (Figure 16b), debris tends to move downstream and accumulate in irregularities on the slope and in couloirs during dry conditions. Later, in wet conditions, they tend to be transported downstream [80]. This rapid mass movement involves debris derived mainly from weathered calcareous and cataclastic bedrock, but also from cemented and uncemented deposits, highlighted by several calcareous boulders and cobbles in a sandy/gravelly matrix (Figure 17).

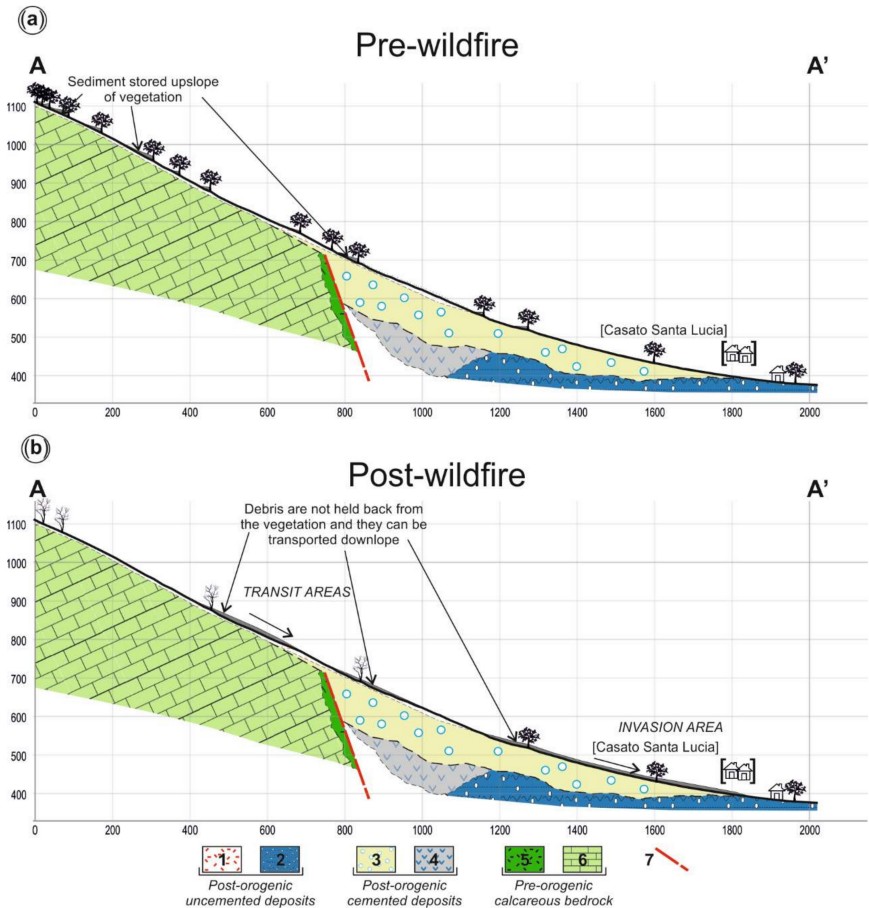

**Figure 16.** Geological cross-section of the pre—(**a**) and post-wildfire (**b**) debris flow channel (cross-section trace in Figure 12 and in Figure 14). Legend: (1) Slope deposits; (2) alluvial plain deposits; (3) alluvial fan deposits; (4) paleo-landslide deposits; (5) cataclastic bedrock; (6) fractured bedrock; (7) normal fault.

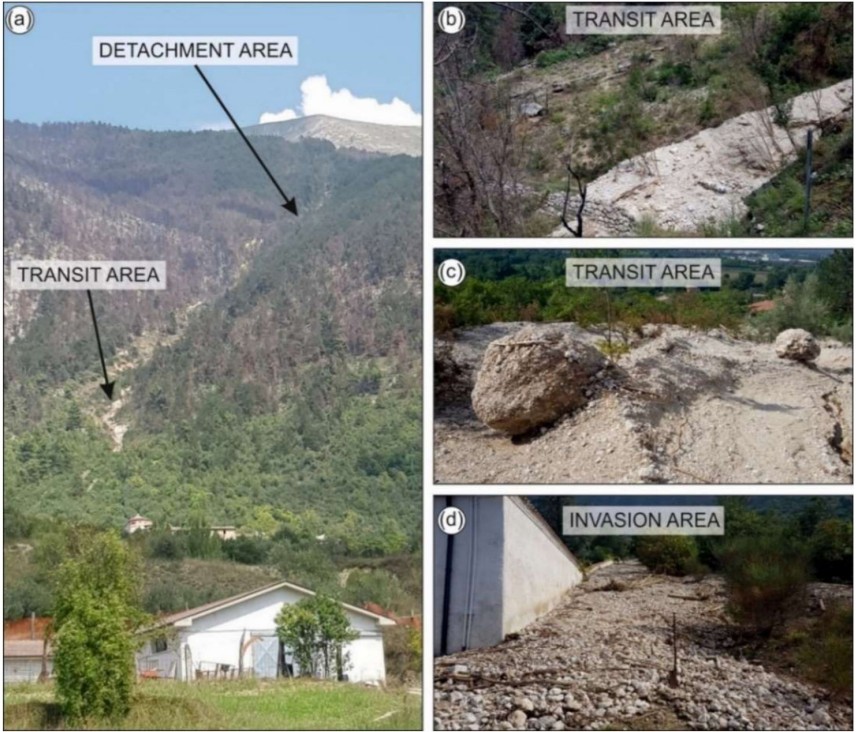

**Figure 17.** (**a**) Panoramic view of the debris flow channel; (**b**–**c**) detail of transit area with the presence of calcareous boulders and cobbles; (**d**) detail of invasion area.

## 5. Discussion

### 5.1. Post-Wildfire Landslide Hazard Mapping

As known in the thematic literature [2,3,81,82] the primary geomorphic consequences of wildfires are dry ravel and root strength decline; furthermore, the decrease in vegetation cover leads to an increase in the infiltration rate that appears to favor the initiation of mass movement, as directly shown in the study area following the occurrence of the August 2018 debris flow.

Post-wildfire landslide hazard mapping was performed though the realization of a geomorphology-based matrix. In detail, the matrix is based on the interaction between factors, which provides certain clues for the post-wildfire hazard assessment, such as morphological features (slope), geological features (lithology and fracturing), geomorphological features (landforms and processes related to gravity and running waters), and post-wildfire vegetation (arboreal and/or shrubby or absent vegetation cover). Each attribute was rated between 1 and 10 according to its criteria and priority, also considering the importance of the factors (Table 3). Each given weight on the attributes was summed altogether and reclassified to define, through a weighted overlay approach [72–75], the spatial relationships between the above-mentioned factors.

The results are graphically shown in the post-wildfire hazard map (Figure 18), in which four classes of landslide hazard (low, moderate, high, and very high) are defined. In particular, the model shows that 20.66% (10.529 km$^2$) and 32.32% (16.474 km$^2$) of the study area were classified as "High" and "Very High" hazard classes, respectively. These areas are located along the steeper scarps and along the main couloir affected by the wildfire. "Moderate" and "Low" hazard classes cover 32.65% (16.644 km$^2$) and 14.37% (7.324 km$^2$) respectively and are located in the upper- and toe-slope, where values of bedrock fracturing, slopes and burned area are extremely lower.

**Table 3.** Factors weight.

| Factors | Attribute | Wt |
|---|---|---|
| Slope(%) | <18 | 1 |
| | 18–45 | 3 |
| | 45–85 | 6 |
| | >85 | 10 |
| Geological features (lithology and fracturing) | Slope deposits | 7 |
| | Alluvial fan deposits | 6 |
| | Paleo-landslide deposits | 2 |
| | Cataclastic bedrock | 10 |
| | Fractured bedrock | 7 |
| | Unfractured bedrock | 1 |
| Geomorphological features (slope gravity landforms and landforms due to running waters) | Active landforms | 9 |
| | Inactive landforms | 6 |
| | Relict landforms | 2 |
| Post-wildfire vegetation cover | Meadow | 5 |
| | Arboreal | 3 |
| | Shrubby or absent | 7 |

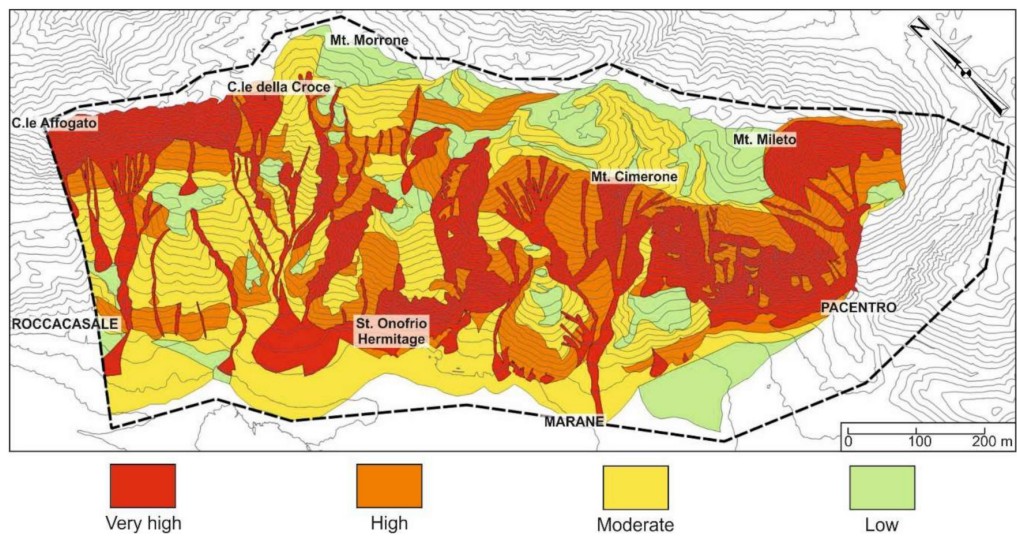

**Figure 18.** Post-wildfire hazard map. The black dashed box indicates the location of the study area.

*5.2. Post-Wildfire Debris Flow Hazard Analysis*

This analysis was performed following a general sequence [79,83] that is defined by the interactions between several factors: Morphometric settings, fire processes, effects on vegetation cover, and rainfall event. Not all burned areas subjected to heavy rainfall produce debris flows, suggesting that some combination of burning conditions, geomorphological landforms, and rainfall characteristics influence susceptibility to debris flows following a fire. First, the morphometric settings of the study area include steep, concave hillslope on a mountainous forested and shrub-dominated environment. Second, a moderate- to high-severity fire (fire processes) and a significant change of the vegetation and lithological properties (fire effects) can highlight the probability of occurrence of this specific mass movement. Next, a rainfall event of enough intensity and duration to produce overland flow (surface runoff) represents the triggering mechanism.

Morphometry exerts strong effects through slope and basic morphometric parameters. Slope value (Figure 7b) is particularly important in relation to the size of contributing areas, with steeper slopes generally requiring smaller upstream areas to generate surface runoff to initiate debris flows than in place where slopes are less steep.

From a morphometric standpoint, the Melton ratio and basin length provide the best identification of basins prone to debris flow in the study area, and their intersection allows a field classification to be delineated [51]. The class limits for the SW slope of the Montagna del Morrone (red line in Figure 19) correctly identified ten (48%) of the 21 field classified basins.

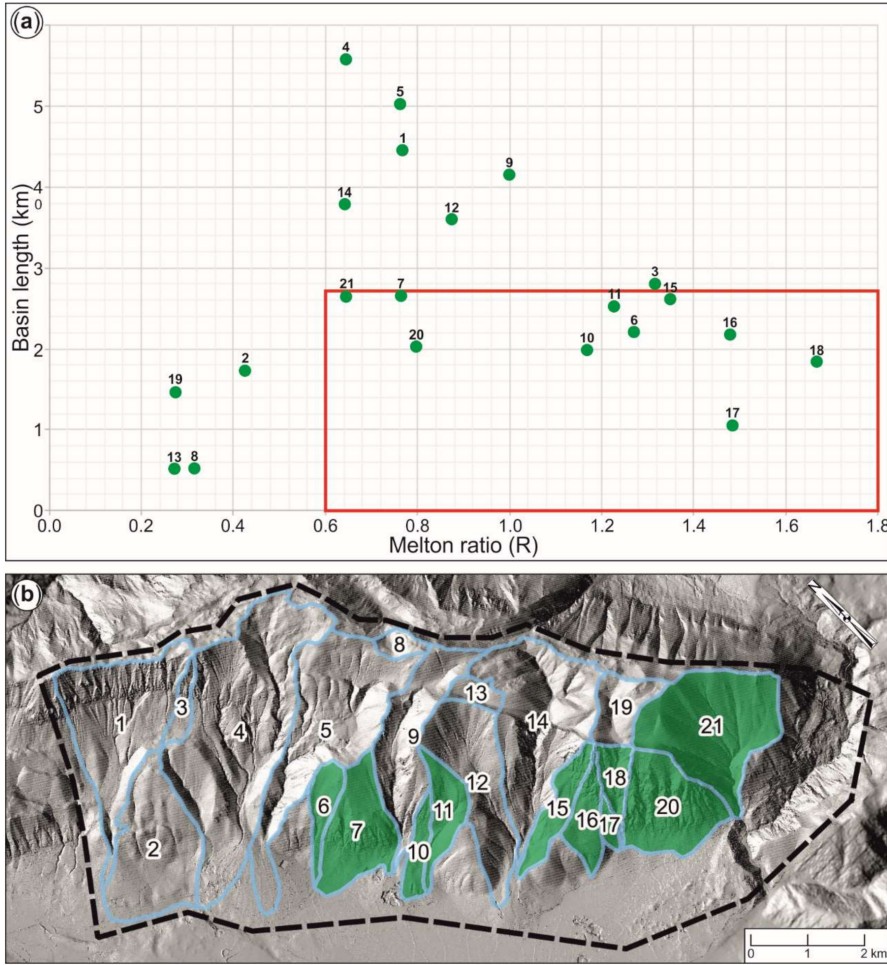

**Figure 19.** (**a**) Scatter diagram using the Melton ratio and basin length with class limits (red line) for the debris flow processes; (**b**) location map of the 21 basins. In blue the basin boundaries; in green basins prone to debris flow according to Wilford et al. [51]. The black dashed box indicates the location of the study area.

The trigger mechanism of the 2018 debris flow event is recognized as being an intense, short duration rainfall event. It occurred from 20:00 on 15 August to 08:00 on 16 August 2018, with a cumulative rainfall of about 90 mm in 12 h. This event took place in the couloir upstream of Casato Santa Lucia, near Marane village, during the vegetation recovery period when rainfall occurring over the disturbed landscape meets little or no resistance compared to vegetated hillslopes. The effect of rainfall on a burned landscape depends on the storm timing relative to the vegetation recovery, intensity, and duration. Canopy loss decreases rainfall interception, regarding both its quantity and its erosive power.

This analysis was applied to basin 11 (Figures 8 and 19b), which presents morphometric features that fit well with the scatter diagram in Figure 19a and was affected by the August 2018 debris flow. This analysis focused on the evaluation of the pre- and post-wildfire setting, highlighting the variation in the contributing area caused by the 2017 fire. The post-wildfire contributing area was evaluated through a detailed geomorphological field survey, supported by a drone survey and integrated with morphometric analysis of the burned area. The result of this evaluation showed an increased variation of 77%, which is graphically shown in Figure 20 and summarized in Table 4.

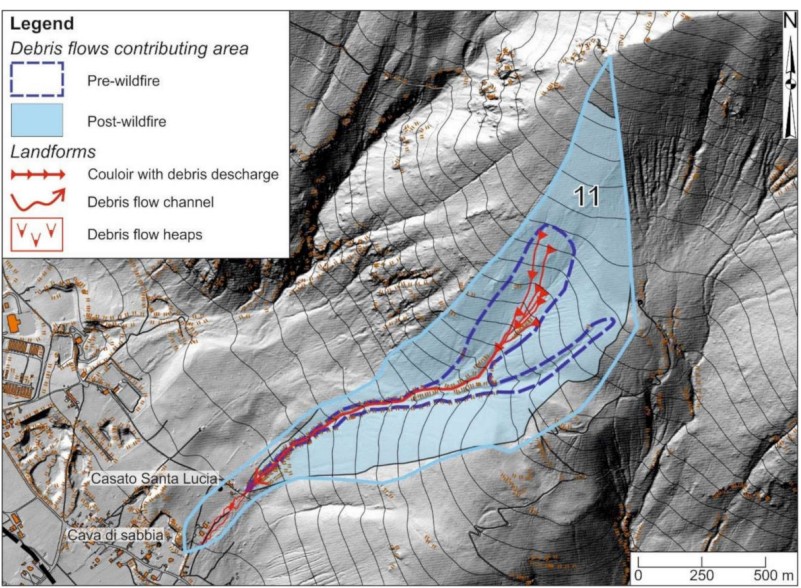

**Figure 20.** Pre- and post-wildfire contributing area for the debris flow event on 16 August. The blue line indicates the watershed of basin 11. For the basin's location see Figures 8 and 19b.

**Table 4.** Dimensions of the debris flow contributing area

| Basin Area (km²) | Pre-Wildfire Contributing Area (km²) | Post-Wildfire Contributing Area (km²) | Area Increased (%) |
|---|---|---|---|
| 0.947 | 0.184 | 0.804 | 77 |

According to the morphometric features delineated in the scatter diagram (Figure 18) and the general lithological and geomorphological setting of the study area, the ten basins prone to debris flows cover a pre-wildfire contributing area of 5.867 km². The similar morphometric and geomorphological post-wildfire setting allowed us to assume the significance of the value of increased contributing area found for basin 11. Hence, this condition led to the analysis being extended to the above-mentioned basins (Figure 19). A post-wildfire contributing area of 10.384 km² was hypothesized.

## 6. Conclusions

This paper presents a post-wildfire landslide hazard assessment realized following a multidisciplinary approach, which involves morphometric analysis, rainfall data analysis, geomorphological field mapping, and geomechanical investigations. This detailed analysis shows clearly that the geomorphological response of the study area is highly variable and dependent on a range of five parameters including slope, post-wildfire vegetation cover, lithological features, bedrock fracturing, and geomorphological elements, which can be considered as instability factors. To better identify these parameters, relevant in the mechanism of landslide occurrence, the analysis was performed following an expert-based weighting procedure, supplemented by GIS software. The aim of this stepwise analysis was to achieve a geomorphology-based matrix, useful to define different

landslide hazard classes. In detail, for the SW slope of the Montagna del Morrone, four classes marking low, moderate, high, and very high landslide hazard were identified.

This work gave a better comprehension of the relations between lithological and geomorphological conditions, and the rainfall events triggering mass movements, such as debris flows; moreover, this analysis was integrated with basic morphometric parameters. In fact, the Melton ratio, in combination with the basin length, was used to recognize basins prone to debris flows. The results of this analysis showed that intense and short duration rainfall can trigger debris flows in ten of the 21 considered basins, identifying a post-wildfire contributing area of ~10 km$^2$. Therefore, the increased probability of debris flows occurrence following wildfire usually lasts for two to three years as documented in Cannon & Gartner [79] and confirmed by the August 2018 debris flow event in the Marane area. After this point, the basins seem to be sufficiently eroded and revegetated so that debris flows are not produced in response to normally occurring rainfall events.

In conclusion, the overall results of this work allowed us to realize a new zonation of landslide hazard for the SW slope of Montagna del Morrone, a mountainous rocky escarpment recently affected by wildfire. This paper provides a methodological contribution to the post-wildfire landslide hazard assessment, particularly for mountainous-forested environments, which can support civil protection warning systems.

**Author Contributions:** Conceptualization, E.M.; Methodology, C.C., E.M. and G.P.; Data Curation, E.M. and N.S.; Geomorphological investigation, GIS mapping and Writing—Original Draft Preparation, C.C. and G.P.; Supervision and Project administration, E.M. and N.S.; Funding Acquisition, E.M. and N.S.

**Funding:** This research and the APC was funded by Nicola Sciarra, grant provided by Abruzzo Region Civil Protection Department and by Enrico Miccadei, grant provided by Università degli Studi dell'Adriatico "G. d'Annunzio" Chieti-Pescara.

**Acknowledgments:** The authors wish to thank the Civil Protection Department and the Abruzzo Civil Protection Office with reference to Sabatino Belmaggio and Ing. Domenico Macrini and the Municipalities of Pacentro, Pratola Peligna, Roccacasale and Sulmona for providing access to the burned areas, logistic and data support. The authors also wish to thank the Cartographic Office of the Abruzzo Region by means of the Open Geodata Portal (http://opendata.regione.abruzzo.it/) and the Ministero dell'Ambiente e della Tutela del Territorio e del Mare (http://www.minambiente.it/), for providing the topographic data, LIDAR, aerial photos and orthophotos used for this work. The rainfall data were provided by the Functional Center and Hydrographic Office of the Abruzzo Region (Centro Funzionale e Ufficio Idrografico Regione Abruzzo). The 20 m SINAnet (Sistema Informativo Nazionale Ambientale) DEM was provided by ISPRA (http://www.sinanet.isprambiente.it/it/sia-ispra/download-mais/dem20/view). The authors wish to thank the anonymous reviewers for their critical review of the paper and their precious suggestions, which significantly improved this manuscript.

**Conflicts of Interest:** The authors declare no conflict of interest.

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
