# Peer review of "Post-Wildfire Landslide Hazard Assessment: The Case of The 2017 Montagna Del Morrone Fire (Central Apennines, Italy)"

_geosciences, doi:10.3390/geosciences9040175_

Round 1

Reviewer 1 Report

Dear authors,

Thank you for your detail study for landslide risk mapping due to the forest fire. The discussion on landslide causative factor maps is very good. However, there are some fundamental aspects related to the aim of the manuscript are not discussed properly. I would like to request you to consider the following comments in the revised version of the manuscript. 

1. First of all, the organization of the manuscript is not appropriate. The tone
of the discussion seems like the literature review. Please reorganize
your manuscript to make it suitable for academic publication. For
instance, Lines 486-489, this kind of statement should be discussed in
the introduction, not in the result section.

"Debris flows are often the most common and most hazardous type of mass movement that can occur after a wildfire. A debris flow can be caused by overland flow generated on moderately to severely burned slopes above a gully or steep couloir; if the flow is much greater than flows that the gully/couloir normally experiences, a debris
flow can result.”

Please consider it an example, there are many examples like the above in the manuscript.

2. You did not highlight any of your interesting result in the abstract, like
the finding of melton ratio-basin length. Your abstract includes only
steps of the research.

3. The conclusion should be more concise and to the point.

4. You have mentioned about landcover change after the fire incidence,
however, you did not analyze any landcover change. Could you please
include landcover maps before and after the forest fire?

5. Your main goal is to map landslide risk. There are no detail methodological
steps for hazard scoring and zonation. You only mentioned about the
heuristic approach but you did not tell the reader what kinds of
heuristics approach you applied. What is the scoring procedure of
different causative factors?? How you incorporate debris flow direction
and speed and transit area. Please add a detail methodological step on
risk mapping in the methodological section. The following reference may
be helpful

Rahman, M. S., Ahmed, B., & Di, L. (2017). Landslide initiation and runout
susceptibility modeling in the context of hill cutting and rapid
urbanization: a combined approach of weights of evidence and spatial
multi-criteria. 
Journal of Mountain Science14(10), 1919-1937.

Van Westen, C. J., Jaiswal, P., Ghosh, S., Martha, T. R., & Kuriakose,
S. L. (2012). Landslide inventory, hazard and risk assessment in India.
In 
Terrigenous Mass Movements (pp. 239-282). Springer, Berlin, Heidelberg.

Lo, C. M., Feng, Z. Y., & Chang, K. T. (2018). Landslide hazard zoning based on numerical simulation and hazard assessment. Geomatics, Natural Hazards and Risk9(1), 368-388.

6. The orientation of some figures is vertical (Fig, 11,13,17). Please
organize these figures in horizontal orientation like other figures.

7. Why you put SRTM 90m DEM in figure 1, while you have LiDAR DEM? Please use LiDAR DEM instead of SRTM.

Author Response

Point 1. First of all, the organization of the manuscript is not appropriate. The tone of the discussion seems like the literature review. Please reorganize your manuscript to make it suitable for academic publication. For instance, Lines 486-489, this kind of statement should be discussed in the introduction, not in the result section.

"Debris flows are often the most common and most hazardous type of mass movement that can occur after a wildfire. A debris flow can be caused by overland flow generated on moderately to severely burned slopes above a gully or steep couloir; if the flow is much greater than flows that the gully/couloir normally experiences, a debris flow can result.”

Please consider it an example, there are many examples like the above in the manuscript.

Response: Ok, we agree with your suggestion. We reorganized the manuscript in order to improve the tone of the discussion modifying the sentence of your example and some other sentences like that. 

Point 2. You did not highlight any of your interesting result in the abstract, like the finding of melton ratio-basin length. Your abstract includes only steps of the research.

Response: Thanks for your suggestion. We reorganized the structure of the abstract highlighting our results.

Point 3. The conclusion should be more concise and to the point.

Response: Ok. All the text was extremally shortened and the conclusion section is now more concise; moreover, we try to be more headed to the point.

Point 4. You have mentioned about landcover change after the fire incidence, however, you did not analyze any landcover change. Could you please include landcover maps before and after the forest fire?

Response: Ok. We insert a pre-wildfire vegetation cover map (figure 5) according to your suggestion.

Point 5. a) Your main goal is to map landslide risk. There are no detail methodological steps for hazard scoring and zonation. You only mentioned about the heuristic approach but you did not tell the reader what kinds of heuristics approach you applied. What is the scoring procedure of different causative factors??

b) How you incorporate debris flow direction and speed and transit area. Please add a detail methodological step on risk mapping in the methodological section.

Response: a) OK we revised the methods section in order to clarify the methodological approach for the hazard evaluation. Post-wildfire landslide hazard analysis was performed following a heuristic or expert-based approach. In this selected method, the type and degree of hazard derive from the geomorphological expert judgment, based on either direct (field mapping) or indirect (GIS data processing) analysis.

b) The debris flow was analyzed through a geomorphological field survey, integrated with a drone survey and mapping. The stepwise approach used to landslide hazard mapping is described in the methodological section and in the previous point (5a).

Point 6. The orientation of some figures is vertical (Fig, 11,13,17). Please organize these figures in horizontal orientation like other figures.

Response: Ok, we organized the figures (also figure 8) in the horizontal orientation and we hope not to lose resolution.

Point 7. Why you put SRTM 90m DEM in figure 1, while you have LiDAR DEM? Please use LiDAR DEM instead of SRTM.

Response: Unfortunately, we don’t have the LiDAR DEM of the entire Central Apennines area. We modified Figure 1 putting a 20m DEM of Italy (SINAnet, ISPRA).

Reviewer 2 Report

see the file

Author Response

Brief summary

The aim of this paper is to provide a thorough study and interpretation of a landslide triggered by heavy rains in an area that had recently undergone severe fires. Detailed analysis of geomorphological, lithological, hydro-meteorological, and geomechanical factors, informed a heuristic landslide analysis of a specific debris flows in the study area. Adapting existing methods and presenting some of their own new approaches, the authors created an updated, post-fire landslide map and offer it to inform civil protection efforts.

Thanks for your comment. In the study we presented a landslide hazard assessment of the entire burned area. The specific debris-flow was used and analyzed as an evidence of the post-wildfire hazard because it is the most evident event mass movement occurred after the wildfire.

Broad comments highlighting areas of strength and weakness.

Overall, this paper demonstrated a multidisciplinary perspective of debris flows, drawing from several well-known methods. It adds to the global body of knowledge in that it investigates and interprets the interactions between fire, vegetation, geomorphology, and rainfall, as they led to a well-documented debris flow. There methodological gaps in some key methods, particularly in how the heuristic map was created and the re-delineation of alimentation area. Given the updated post-fire landslide hazard map, the authors do not venture to discuss why a debris flow occurred in Basin 11 and not others, even though many of the explanatory factors (landslide matrix) are similar. These weaken the authors’ proposed contributions to civil protection warning and general post-fire landslide analysis. Further clarification of specific methods as noted in my specific comments could address these weaknesses.

Thanks for your comment. We rewrote some paragraphs, especially methods and discussion section in order to be clearer in our study.

Specific comments referring to line numbers, tables or figures

1.       Line 188. Very briefly describe what approach Frate et al. used to classify fire severity. For instance, did they use remote sensing (and if so what spectra or indices), or field surveys?

Response: Ok, we explained in few sentences how Frate et al. determined fire severity. In particular, they defined the severity classes using pre and post Sentinel 2-A data of the burned areas, from which the Normalized Burned Ratio Index [Key & Benson 2006] was obtained.  

2.       Figure 4. It would greatly help to see both panels oriented in the same cardinal direction.

Response: Ok, done. Thanks for your suggestion

3.       Line 219. Clarify how vegetation cover was reclassified.

Response: We reclassified the vegetation cover grouping beeches and conifers into arboreal vegetation cover and shrublands and absent in shrubby or absent vegetation cover. Furthermore, in the post-wildfire vegetation cover map we considered all the burned area as a shrubby or absent vegetation cover area.

4.       Figure 5. It is not clear if this vegetation map represents pre- or post-fire conditions.  If possible, side-by-side pre- and post-fire vegetation maps would greatly complement the discussion on fire’s impact on vegetation and ultimately slope stability and could relate well to Figure 15.

Response: Ok. We inserted a pre-wildfire vegetation cover map (Figure 5) and a post-wildfire vegetation cover map (Figure 6) according to your suggestion.

5.       Line 238. Specify how you decided to classify into 21 basins, or if this was a pre- determined delineation.

Response: Methods section was integrated. In particular, basin boundaries and drainage lines were digitized from 1:5,000 topographic maps and verified by means of 1:33,000 and 1:10,000 air-photos. The study area was classified into 17 basins, that include at least a second order stream, and four areas grouped into “hydrographic units”, which show a homogeneous distribution of first order stream, gullies and channels along steep to sub-vertical scarps. This analysis led to define a total of 21 basins, of which basic morphometric parameters (such as area, perimeter, relief, length) were obtained from the DEM.

6.       Line 282. Correct spelling of “software”

Response: Ok, done.

7.       Section 4.3 This section and subsections 4.3.1, 4.3.2, 4.3.3, are too standalone. It would help to discuss this relevant to fires, not just landslides.

Response: Thanks for your suggestion. These sections have been inserted to introduce the role of the lithologies in the landslides triggering mechanism, which was hardly modified by the fire occurrence. Furthermore, by entering the description of the weights in the discussion section the role of the lithologies is clearer and better connected to the rest of the manuscript.

8.       Line 436 / Figure 11. Make these locations more obvious in Figure 11. It is very hard to find these in the figure.

Response: Ok, done.

9.       Figure 14. Add a location map so that the readers very clearly see where this debris flow occurred.

Response: Thanks for your suggestion. Done.

10.   Figure 4, 5, 6, 7, 9, 11, 13. Spatial context is very important with the arguments and interpretations the authors are making. Put an overlay of the specific debris flows area (from Fig 14) throughout figures in this paper.

Response: Ok. As noted above, we presented a landslide hazard assessment of the entire burned area. The specific debris-flow was used and analyzed as an evidence of the post-wildfire hazard. In our opinion is not necessary to put the location of the debris flow in all the suggested figures; furthermore, in the geolithological (fig. 12) and geomorphological (fig. 14) maps the cross-section trace matches with the debris flow channel. We added also a clarification in the figure caption (figure 12 and 14).

11.   Line 539. “Used previously.” Authors either need to a) provide a more explicit description of the methods on how each parameter was reclassified and ultimately combined to create the final heuristic map, or b) more clearly state exactly who else’s method was replicated.

Response: Ok. We largely reorganized the method and discussion section, clarifying the methodology used to classify each parameter and combine them in the final map with a weighted overlay process.

12.   Line 543. Need to specify if pre- or post-fire vegetation.

Response: Ok, done. Thanks for your suggestion

13.   Section 5.2 (Lines 551-582). Same comment as Line 539. Are these definitions novel to this paper, or are they replicating another method? If the former, provide further justification on why these ranges. If the latter, clearly cite whose method is being applied.

Response: As noted above on point 11 we largely reorganized the discussion section.

14.   Lines 609-611. The authors have not provided justification for this statement.

Response: Thanks for this comment! Our mistake; there was a problem with the reference to Wilford el alii 2004.

15.   Line 628. Be consistent in using quantity of rainfall and duration of rainfall, throughout the manuscript.

Response: We uniformed this point throughout all the manuscript according to your suggestion. In detail, as shown in subsection 4.2.1, at 20.00 of 15 August the cumulative rainfall was 9.6 mm, while at 08.00 of 16 August was 96.2 mm; so, the cumulative rainfall of the extreme event was 86.6 mm in 12h so in the manuscript we wrote about 90 mm.

16.   Line 636. Alimentation is a key idea in the development of the novel aspects of this paper, but it lacks a clear description early on in the manuscript. It is not clear how the authors justified their delineations of pre- and post- fire alimentation area. Provide more background and justification on alimentation area earlier and throughout the paper.

Response: Post-wildfire contributing area was evaluated thought a detailed geomorphological field survey, supported by a drone survey and integrated with morphometric analysis of the burned area. We provided background and justification either in the method and discussion section.

17.   Line 684. This study did not prove this point, but maybe it added additional evidence to Cannon and Gartner 2005. Re-word this sentence accordingly.

Response: Ok sentence revised.

Reviewer 3 Report

Overall:

This is an interesting paper.  In its present state it is rather wordy and difficult to read.  It can be improved greatly by a thorough revision for good English grammar. In addition, it needs to be reduced in size and this can be done by removing excess words.  In other words, make this paper much more terse.

To accomplish this, I suggest the authors focus on the landslide modeling.  The introductory material regarding the geology of the area, etc needs to be shortened substantially.  I would expect this paper --in final form-- should be no more than 20 pages in length (it is currently 31).

Abstract:

The abstract should be polished for improved English grammar.  For example, the first sentence should be "This work focused on a post-wildfire..."

Similar improvements would greatly improve the read-ability of this paper.

Line 74-78, one huge run-on sentence. In fact the entire paragraph is one sentence.  This needs to be re-written to improve read-ability.

Line 188: How was fire severity determined.  Summarize in a sentence or two what Frate et al did.

The conclusions section is very long.  It should be just a few paragraphs.  Furthermore, after reading it twice, I still do not know what the real conclusions are.  The authors need to re-think this and write a very solid "take home" message to the reader.  The authors alllude to various factors important in determining landslide risk but these important factors are not made prominent in the conclusions.

Author Response

Overall:

This is an interesting paper. In its present state it is rather wordy and difficult to read. It can be improved greatly by a thorough revision for good English grammar. In addition, it needs to be reduced in size and this can be done by removing excess words.  In other words, make this paper much more terse. To accomplish this, I suggest the authors focus on the landslide modelling. The introductory material regarding the geology of the area, etc needs to be shortened substantially.  I would expect this paper --in final form-- should be no more than 20 pages in length (it is currently 31).

Response: Thanks for your suggestion, we have checked the text again polishing it and a further English grammar and syntax was made. We tried also to short the text focusing on our goal and now the manuscript is of 23 pages (excluding the references).

Abstract:

The abstract should be polished for improved English grammar.  For example, the first sentence should be "This work focused on a post-wildfire..." Similar improvements would greatly improve the read-ability of this paper.

Response: Thanks for your suggestion, all the text was largely revised and reorganized in order to improve the readability of the paper. Moreover, the manuscript underwent a check by the MPDI English Editing Service for language review and a final check of the English language will be done at the end of the revision process.

Line 74-78, one huge run-on sentence. In fact the entire paragraph is one sentence. This needs to be re-written to improve read-ability.

Response: Ok, sentence revised.

Line 188: How was fire severity determined. Summarize in a sentence or two what Frate et al did.

Response: Ok, we explained in one sentence how Frate et al. determined fire severity. In particular, they defined the severity classes using pre and post Sentinel 2-A data of the burned areas, from which the Normalized Burned Ratio Index [Key & Benson 2006] was obtained.  

The conclusions section is very long. It should be just a few paragraphs. Furthermore, after reading it twice, I still do not know what the real conclusions are. The authors need to re-think this and write a very solid "take home" message to the reader. The authors allude to various factors important in determining landslide risk but these important factors are not made prominent in the conclusions.

Response: Ok, we agree with this comment. We shortened this section in order to write the real conclusion of this paper in a concise way. Moreover, we reorganized the section underlining the methodology and the factors important in determining landslide hazard.  

Round 2

Reviewer 1 Report

Dear Authors

Thank you for the revised version. However, I think the revised version did not address all the issues properly. 

I did not find any result in the abstract.

I did not find any scoring procedure of AHP for hazard scoring. your main methodology is very unclear.

Author Response

Reviewer 1 – round2

Dear Authors

Thank you for the revised version. However, I think the revised version did not address all the issues properly. 

I did not find any result in the abstract.

I did not find any scoring procedure of AHP for hazard scoring. your main methodology is very unclear.

Response: Ok. We largely reorganized the methods and discussion section. We clarified the main methodology used to analyze the instability factors according to the expert-based weighted overlay process. Furthermore, we added more information about our results in the abstract section.

Reviewer 2 Report

The authors made the suggested edits to clarify their methods, presentation of figures, and made clearer references to other papers.

Author Response

Thanks for approving our modifications. We made a further revision of the manuscript according to the new suggestions, in order to more clearly describe especially the methods and to improve the overall description of our research.

Reviewer 3 Report

I believe this ms is now ready for publication. It is still a bit on the lengthy side but that is really  the call of the Editor at this point.

Author Response

Thanks for approving our modifications and suggesting the revised version of the manuscript for pubblication. A futher check of the text was made, focused typos and language - style refinements.

Round 3

Reviewer 1 Report

Dear Authors 

Thank you for your effort to revised the paper. Perhaps I failed to make my comments clear enough. Please consider the following comments carefully for your next review.

You have mentioned in your abstract 

"....recognize basins prone to debris flows through basic morphometric parameters, such as Melton ratio in combination with basin length"

could you please tell the reader what is the relation between Melton ratio and basin length in an abstract more specifically.

2.  Secondly, can you please add a table of expert-based weighted overlay? The readers may like to read the weight of different factors. We may learn from the score which factor is more important than others. Please have a look at the following references:

Saaty, R. W. (1987). The analytic hierarchy process—what it is and how it is used. Mathematical modelling9(3-5), 161-176. 

The above reference discussed how to determine weight and present different factor maps based

Rahman, M. S., Ahmed, B., & Di, L. (2017). Landslide initiation and runout susceptibility modeling in the context of hill cutting and rapid urbanization: a combined approach of weights of evidence and spatial multi-criteria. Journal of Mountain Science14(10), 1919-1937. 

Please have a look from table 2 to table 6 for the examples  of the weight table

Author Response

Thank you for your effort to revised the paper. Perhaps I failed to make my comments clear enough. Please consider the following comments carefully for your next review.

POINT 1

You have mentioned in your abstract 

"....recognize basins prone to debris flows through basic morphometric parameters, such as Melton ratio in combination with basin length"

could you please tell the reader what is the relation between Melton ratio and basin length in an abstract more specifically.

Response: Thanks for your suggestion. In the abstract we specified the class of basins prone to debris flows with Melton ratio values >0.6 and basin length values <2.7, according to Wilford et al. 2004. In the methods and discussion sections this point is well described.

POINT 2

Secondly, can you please add a table of expert-based weighted overlay? The readers may like to read the weight of different factors. We may learn from the score which factor is more important than others. Please have a look at the following references:

Saaty, R. W. (1987). The analytic hierarchy process—what it is and how it is used. Mathematical modelling9(3-5), 161-176. 

The above reference discussed how to determine weight and present different factor maps based

Rahman, M. S., Ahmed, B., & Di, L. (2017). Landslide initiation and runout susceptibility modeling in the context of hill cutting and rapid urbanization: a combined approach of weights of evidence and spatial multi-criteria. Journal of Mountain Science14(10), 1919-1937. 

Please have a look from table 2 to table 6 for the examples of the weight table

Response: Thanks for suggesting us these specific references, which were largely analyzed. In this study each attribute, summarized in Table 3, was rated between 1 and 10 considering the importance of the five factors in landslides occurrence, and according to its criteria and priority, defined by expert opinion. This aspect is clarified in the method section. The expert-based weighed overlay process presented in this paper, is performed according to the following reference:

Roslee, R.; Mickey, A.C.; Simon, N.; Norhisham, M.N. Landslide susceptibility analysis (LSA) using weighted overlay method (WOM) along the Genting Sempah to Bentong highway, Pahang. Malaysian J. Geosci. 2018. doi: 10.26480/mjg.02.2017.13.19

Furthermore, your suggested references are really interesting and relevant, and suggest a significant methodology, useful to realize landslide hazard and susceptibility maps. For this reason, the references will be taken into account for our future papers concerning these issues applied in similar contexts.
